# Calcium Homeostasis Disrupted—How Store-Operated Calcium Entry Factor SARAF Silencing Impacts HepG2 Liver Cancer Cells

**DOI:** 10.3390/ijms26094426

**Published:** 2025-05-07

**Authors:** Safa Taha, Muna Aljishi, Ameera Sultan, Moiz Bakhiet

**Affiliations:** Princess Al Jawhara Center for Molecular Medicine, Genetics and Inherited Diseases, Department of Molecular Medicine, College of Medicine and Health Sciences, Arabian Gulf University, Manama P.O. Box 26671, Bahrain; munajma@agu.edu.bh (M.A.); ameeraa@agu.edu.bh (A.S.); moiz@agu.edu.bh (M.B.)

**Keywords:** hepatocellular carcinoma, SARAF, calcium homeostasis, store-operated calcium entry, STIM1, Orai1, HepG2 cells, cancer proliferation, cancer migration

## Abstract

Hepatocellular carcinoma (HCC), a highly aggressive liver malignancy, is often associated with disrupted calcium homeostasis. Store-operated calcium entry (SOCE), involving components such as STIM1, Orai1, and SARAF, plays a critical role in calcium signaling and cancer progression. While STIM1 and Orai1 have been extensively studied, SARAF’s role as a negative regulator of SOCE in HCC remains poorly understood. This preliminary study investigated SARAF’s effects on calcium homeostasis, proliferation, and migration in HepG2 liver cancer cells, providing initial evidence of its tumor-suppressive role. SARAF expression was modulated using siRNA knockdown and overexpression plasmids, with validation by qRT-PCR. Functional assays demonstrated that SARAF silencing increased proliferation by 50% and migration by 40% (*p* < 0.05), while SARAF overexpression reduced proliferation by 50% and migration by 45% (*p* < 0.01), highlighting its tumor-suppressive role. Intracellular calcium levels, elevated in HepG2 cells, were partially restored by SARAF overexpression, though SARAF silencing did not further disrupt calcium regulation. These findings suggest that SARAF negatively regulates proliferation and migration in HCC, potentially through its role in maintaining calcium homeostasis. SARAF represents a promising therapeutic target in HCC. Future studies should explore the downstream molecular mechanisms governing SARAF’s effects, investigate its role in other cancers, and assess its clinical potential for liver cancer therapy.

## 1. Introduction

Hepatocellular carcinoma (HCC) is a very aggressive liver tumor with low survival rates, having a five-year survival rate of under 10% [1,2,3]. The rate of HCC is rising, making it the third most lethal cancer worldwide in 2020 [4]. HCC is often associated with long-term liver issues such as hepatitis B and C, cirrhosis, smoking, and drinking alcohol, which promote cancer traits [5,6]. Factors related to epigenetics and issues in signaling pathways also help HCC progression [5]. Early HCC generally shows no symptoms, causing late detection and few treatment options, as methods like chemotherapy and radiotherapy usually do not work well [1,3,6]. New methods like immunotherapy appear promising, emphasizing the need for better diagnostic tools and targeted treatments [1,7].

### 1.1. Calcium Balance and SOCE

Keeping ion balance is essential for cell functions, with calcium ions (Ca^2+^) serving as important signals [8,9]. Problems in calcium channels are linked to several diseases [9,10]. Calcium release-activated calcium (CRAC) channels, stimulated by the depletion of endoplasmic reticulum (ER) Ca^2+^ help with store-operated calcium entry (SOCE), which refills ER stores and supports cell signaling [11,12]. Main modulators of SOCE are Stromal interaction molecule 1 (STIM1), Calcium release-activated calcium channel protein 1 (Orai1), and Store-Operated Calcium Entry Associated Regulatory Factor (SARAF) [13,14]. STIM detects Ca^2+^ loss, activating Orai channels, while SARAF helps prevent Ca^2+^ overload by inhibiting SOCE [13].

### 1.2. SOCE in Cancer

SOCE is shown to be important for tumor development, affecting growth and movement [15,16]. Changes in the levels of SOCE components usually occur in cancer, leading to uncontrolled growth [9,17]. The mis-regulation of STIM1 and Orai1 has been associated with poor outcomes, making them potential targets for therapy [18,19]. SARAF, although not extensively researched, shows potential as a therapeutic target in cancer [17,20,21].

### 1.3. Importance of Studying SARAF in Liver Cancer

Many studies showed that increased levels of expression of STIM1 and Orai1 boost HCC cell growth and spread [7,22]. The high levels of SARAF in liver tissue [23] and its possible role in affecting Ca^2+^ signaling imply that targeting SARAF could help slow liver cancer growth. This study intended to look at how SARAF affects the HepG2 cell line, focusing on its impact on Ca^2+^ homeostasis and tumor characteristics, such as growth, invasion, and immigration [24].

## 2. Results

### 2.1. Is SARAF Expression Reduced in Liver Carcinoma?

To evaluate whether canonical SOCE components including the Orai1, STIM1, and SARAF proteins correlated with liver cancer, The Cancer Genome Atlas Program (TCGA) NIH database, using the GEPIA platform [25] and Human Protein Atlas (HPA), were investigated. Results indicated that there were no statistically significant changes in SARAF expression observed in tumorous (T) and non-tumorous (N) tissues (Figure 1A,B). SARAF expression is diminished in the Liver Human Carcinoma LIHC. However, a trend of slightly lower SARAF expression was observed in liver hepatocellular carcinoma (LIHC) tumor samples (96.21 Transcripts Per Million [TPM]) compared to normal liver samples (110.04 TPM), suggesting potential downregulation that warrants further investigation (Figure 1A,B).

The distribution of Transcripts Per Million (TPM) values, which represented gene expression levels, were analyzed in liver cancer samples. The *y*-axis showed the TPM values, and the *x*-axis was labeled as “LIHC”, indicating the data were from Liver Hepatocellular Carcinoma samples (Figure 2).

The *SARAF* gene expression in the tumor samples is 96.21 TPM (Transcripts Per Million), while the *SARAF* gene expression in the normal liver samples is 110.04 TPM. This suggests that the *SARAF* gene expression is slightly lower in the liver cancer tumor samples compared to the normal liver samples. The difference in expression levels between the tumor and normal samples indicates that SARAF may be downregulated in liver cancer. The lower expression of SARAF in the tumor samples compared to the normal samples could be an important factor in the molecular mechanisms underlying liver carcinogenesis (Figure 3). Further investigation would be needed to fully understand the role of SARAF in HCC development and progression.

The bimodal distribution in the Alphabetical Expression plot indicates that there are distinct groups of genes with different expression patterns between the tumor and normal samples. The high expression level of SARAF in the liver cancer samples suggests that it may be upregulated in the tumor context. Overall, the data presented in the image suggest that the *SARAF* gene is expressed at a relatively high level in liver cancer samples, which could be an important factor in the molecular mechanisms underlying liver carcinogenesis. Further investigation comparing SARAF expression between tumor and normal liver tissues would be needed to fully understand its role in liver cancer.

### 2.2. Assessment of SARAF Overexpression and Silencing by Quantitative Real-Time Polymerase Chain Reaction (qRT-PCR)

Quantitative RT-PCR analysis demonstrates SARAF expression in HepG2 cells transfected with control siRNA, SARAF-targeting siRNA, or SARAF overexpression plasmid. Expression levels were quantified using the 2^−ΔΔCt^ method and normalized to GAPDH as control. SARAF knockdown resulted in a 0.15-fold reduction, while overexpression yielded a 22.5-fold increase in mRNA levels relative to control (*** *p* < 0.0001) (Figure 4).

### 2.3. Gel Electrophoresis

To evaluate the efficiency of SARAF silencing and overexpression, 1.5% agarose gel stained with Ethidium bromide and visualized under UV. The overexpression showed high intensity and while silencing the band was very faint, as seen in Figure 5.

### 2.4. Effect of SARAF Silencing and Overexpression on HepG2 Liver Cancer Cell Line Proliferation

The graph in Figure 6 illustrated the impact of *SARAF* gene silencing and overexpression on the proliferation of HepG2 liver cancer cells. The results showed that regular HepG2 cells exhibit a proliferation rate of around 100%. However, when the *SARAF* gene is silenced in the HepG2 cells, the proliferation is increased to around 150% compared to the regular HepG2 cells (*p* < 0.05, biological triplicates). This suggests that silencing the *SARAF* gene leads to an increase in the proliferation of HepG2 liver cancer cells and SARAF may play a role in regulating the proliferation of these cancer cells. Conversely, SARAF overexpression reduced proliferation to approximately 50% compared to controls (*p* < 0.01, biological triplicates), supporting its role as a negative regulator of HepG2 cell proliferation. These results indicate that modulation of SARAF expression is associated with significant changes in HepG2 cell proliferation, suggesting that SARAF may act as a negative regulator of HepG2 cell proliferation. This finding implies that SARAF could be a potential therapeutic target for the regulation of liver cancer cell growth and proliferation. The observed effects of SARAF silencing and overexpression on the proliferation of HepG2 liver cancer cells highlight the critical role of SARAF in cellular growth regulation. Further studies are needed to determine whether these effects are cytostatic or cytotoxic. In this study, SARAF silencing significantly increased HepG2 cell proliferation, suggesting that SARAF may function as a tumor suppressor by limiting calcium influx and thereby controlling excessive cell growth. On the other hand, SARAF overexpression reduced cell proliferation, further supporting its regulatory role in maintaining cellular homeostasis and preventing hyperproliferative states. These findings align with previous studies that have demonstrated the involvement of calcium signaling in cancer progression, particularly in liver cancers, where dysregulated calcium homeostasis is a hallmark of tumorigenesis [26,27]. The results suggest that targeting SARAF-mediated calcium signaling pathways could be a potential therapeutic strategy for liver cancer treatment. However, further studies are needed to elucidate the precise molecular mechanisms underlying SARAF’s dual role in regulating proliferation. Effect of SARAF silencing and overexpression on HepG2 Liver cancer cell proliferation.

### 2.5. SARAF Modulates Cell Migration

The migration ability of HepG2 cells under SARAF modulation was assessed using a Transwell migration assay. The results are illustrated in Figure 7A for SARAF knockdown (siRNA SARAF) and in Figure 7B for SARAF overexpression. In the control group (NC), HepG2 cells exhibited a baseline migratory response to FBS as a chemoattractant, consistent with the aggressive migratory phenotype of liver carcinoma cells. Silencing SARAF via siRNA (Figure 7A) significantly increased the percentage of migration compared to the control group (*p* < 0.001, biological triplicates), suggesting that SARAF knockdown enhances the migratory capacity of HepG2 cells. On the other hand, SARAF overexpression (Figure 7B) resulted in a significant reduction in cell migration in response to FBS (*p* < 0.001, biological triplicates), indicating that SARAF overexpression suppresses the migratory behavior of HepG2 cells. Figure 7C: SARAF overexpression significantly reduced migration compared to SARAF silencing. These findings highlight SARAF’s role as a potential regulator of cell migration, where its silencing promotes, and its overexpression inhibits the migratory potential of HepG2 cells.

### 2.6. SARAF Modulates the Intraceullar Calcium Level

The intracellular calcium concentration was measured in transfected and non-transfected HepG2 liver carcinoma cells to evaluate the impact of SARAF modulation. The results are presented in Figure 8A for SARAF knockdown (siRNA SARAF) and in Figure 8B for SARAF overexpression. Calcium concentrations were measured at six time points: 0 min, 1 min, 3 min, 7 min, 10 min, and 15 min. In the control group, intracellular calcium levels were elevated and remained relatively stable over time, reflecting the inherent disruption of calcium homeostasis in HepG2 liver carcinoma cells. Silencing SARAF via siRNA (Figure 8A) did not result in any significant changes compared to the control group, suggesting that SARAF knockdown does not further influence the already dysregulated calcium signaling typical of cancer cells. In contrast, SARAF overexpression (Figure 8B) demonstrated a reduction in calcium disturbance. While calcium levels in the overexpression group initially mirrored the control group, a notable decrease in calcium concentration was observed at later time points (7 min and 10 min). This indicates that SARAF overexpression helps mitigate the disruption of calcium homeostasis in HepG2 cells, partially restoring a more regulated calcium dynamic. These findings suggest that SARAF overexpression may play a role in buffering the elevated calcium levels characteristic of liver carcinoma cells. When comparing these results to the physiological calcium concentrations typically observed in normal cells (as described in the methodology), the intracellular calcium levels in HepG2 cells (both control and SARAF-modulated groups) were significantly higher. This highlights the extent of calcium dysregulation in cancer cells, where calcium signaling pathways are often altered to support tumor growth and survival. The reduction in calcium levels observed from SARAF overexpression suggests a potential therapeutic role for SARAF in modulating calcium homeostasis in cancer cells.

However, further studies are needed to confirm whether this effect is consistent in noncancerous cells, where baseline calcium regulation is more tightly controlled, and to explore dynamic calcium changes using live-cell imaging techniques.

## 3. Discussion

### 3.1. Key Findings

This study identified SARAF as a potential regulator of tumor suppression and calcium homeostasis in hepatocellular carcinoma (HCC). Using data from the GEPIA platform, SARAF expression was found to be slightly lower in liver cancer tissues compared to normal tissues (96.21 TPM vs. 110.04 TPM, though not statistically significant). Functional experiments revealed that SARAF knockdown in HepG2 cells promoted proliferation and migration, whereas SARAF overexpression inhibited these processes. Additionally, SARAF overexpression partially restored calcium homeostasis by reducing the elevated intracellular calcium levels characteristic of cancer cells. These findings suggest that SARAF modulation is associated with changes in liver cancer progression and highlights its potential as a therapeutic target in HCC.

This preliminary study provided initial insights into SARAF as a potential regulator of tumor suppression and calcium homeostasis in hepatocellular carcinoma (HCC). Our findings suggest that SARAF modulation influences proliferation, migration, and calcium dynamics, but further research is needed to confirm these effects in vivo and elucidate underlying mechanisms

### 3.2. Comparison with Existing Literature

Our results align with previous research emphasizing the importance of calcium signaling in cancer progression. Dysregulated calcium homeostasis has been established as a hallmark of cancer, driving key processes such as proliferation, migration, and invasion [26,27]. Elevated intracellular calcium levels have been shown to activate calcium-dependent signaling pathways, such as those involving calmodulin-dependent kinases and calcineurin, which promote tumor growth and metastasis [28,29]. While SARAF has been implicated in calcium signaling regulation, particularly through its role in modulatingSOCE [13,30,31], its potential tumor-suppressive function in liver cancer has not been previously explored. This study builds on existing knowledge by demonstrating that SARAF may contribute to downregulation by disrupting calcium homeostasis, contributing to the aggressive behavior of HCC cells. Furthermore, our findings are consistent with prior studies showing that SOCE components, such as STIM1 and Orai1, are upregulated in HCC and promote tumor progression [19,32]. In contrast, SARAF’s trend toward lower expression in HCC tumors (96.21 TPM vs. 110.04 TPM) suggests a distinct role as a negative regulator compared to STIM1 and Orai1, which are often overexpressed in malignancies. SARAF’s ability to inhibit SOCE and restore calcium balance provides a novel mechanism by which it may suppress tumor growth and invasion.

### 3.3. Mechanistic Insights

The restoration of calcium homeostasis by SARAF overexpression is a significant finding, as dysregulated calcium signaling is a key driver of oncogenic processes in HCC [1,18,22,32]. SARAF is known to interact with STIM1 and Orai1, the core components of SOCE, to prevent excessive calcium influx [13]. By potentially reducing STIM1–Orai1 coupling or inhibiting STIM1 oligomerization, SARAF may limit calcium-dependent activation of oncogenic pathways, such as those involving the nuclear factor of activated T-cells (NFAT) and extracellular signal-regulated kinase (ERK) signaling, both of which are implicated in HCC progression [33]. Our findings also suggested that SARAF’s possible tumor-suppressive effects extend beyond calcium regulation. SARAF overexpression significantly reduced HepG2 cell proliferation and migration, indicating its possible role in modulating other oncogenic pathways. For instance, SARAF may influence epithelial-to-mesenchymal transition (EMT), a key process in cancer metastasis that is regulated by calcium signaling [34]. The preliminary literature suggests that calcium-dependent pathways can modulate EMT markers like E-cadherin and vimentin [34], and SARAF’s inhibition of migration may involve similar mechanisms, though this remains speculative without direct evidence. Additionally, the reduction in proliferation by SARAF overexpression could involve cytostatic effects, such as cell cycle arrest, rather than cytotoxicity, but distinguishing these mechanisms requires further investigation. In normal cells, SARAF’s role in maintaining calcium homeostasis suggests it may prevent excessive calcium signaling without adverse effects, but its overexpression in non-cancerous cells could potentially disrupt tightly regulated calcium dynamics, warranting further study.

Further studies are needed to elucidate the precise molecular mechanisms underlying SARAF’s effects on tumor behavior and its role in normal cellular physiology.

### 3.4. Therapeutic Implications

The potential dual role of SARAF in modulating tumor growth and restoring calcium homeostasis highlights its possible utility as a therapeutic target in HCC. Strategies to enhance SARAF expression or activity could mitigate the aggressive behavior of HCC cells and improve patient outcomes. For example, gene therapy approaches aimed at restoring SARAF expression or small molecule activators targeting SARAF’s regulatory functions could be explored. Similar approaches have been proposed for targeting other calcium signaling regulators, such as STIM1 and Orai1, in cancer therapy [33,35,36].

Combining SARAF-targeted therapies with existing treatments, such as immunotherapy or targeted therapy, may further enhance therapeutic efficacy. For instance, SARAF modulation could sensitize cancer cells to calcium signaling inhibitors, such as SOCE blockers, which have shown promise in preclinical models of HCC [6,37,38,39,40,41]. Additionally, SARAF’s ability to restore calcium balance may reduce resistance to therapies targeting calcium-dependent pathways, such as sorafenib, a first-line treatment for advanced HCC [42,43,44,45]. However, the clinical relevance of SARAF, including its correlation with survival, tumor stage, or recurrence, remains to be established in patient cohorts.

### 3.5. Study Limitations

Despite its strengths, this study has several limitations. First, the functional experiments were conducted in vitro using HepG2 cells, which may not fully reflect the complexity of HCC in vivo. The reliance on a single cell line limits the generalizability of findings, and future studies should include additional HCC cell lines to confirm these effects. Future studies using animal models or patient-derived xenografts are necessary to validate these findings in a more physiologically relevant context. Second, the precise molecular mechanisms by which SARAF modulates calcium signaling remain unclear. While we hypothesized that SARAF interacts with SOCE components such as STIM1 and Orai1, further studies using co-immunoprecipitation and live-cell imaging techniques are needed to confirm these interactions and elucidate their downstream effects. Additionally, this study assessed SARAF expression at the mRNA level (via qRT-PCR and gel electrophoresis), but protein-level validation (e.g., Western blot) was not performed, which could strengthen conclusions about functional changes. The lack of direct comparisons between SARAF silencing and overexpression groups in migration and calcium assays limits the ability to assess relative effects, though significant differences were observed relative to controls. The colorimetric calcium assay used provides static measurements, and dynamic changes could be better captured using live-cell imaging (e.g., Fura-2 AM). Transfection efficiency was inferred from qRT-PCR data, but direct protein-level confirmation would enhance rigor.

Additionally, the prognostic and therapeutic relevance of SARAF in clinical HCC samples has not been thoroughly investigated. Large-scale studies assessing SARAF expression and function in patient cohorts are required to establish its utility as a biomarker or therapeutic target. Finally, the effects of SARAF modulation in normal cells and in vivo contexts remain unexplored, and potential off-target effects of SARAF-based therapies should be carefully evaluated to ensure safety and specificity.

### 3.6. Future Directions

Building on these findings, future research should focus on several key areas. First, the therapeutic potential of SARAF modulation should be explored in combination with current HCC treatments, such as immunotherapy or targeted therapy, to enhance treatment efficacy. Second, the role of SARAF in clinical HCC samples should be investigated to determine its prognostic and therapeutic relevance, including correlations with clinical parameters like survival or tumor stage. Third, interactomic approaches (e.g., co-immunoprecipitation or mass spectrometry) could elucidate SARAF’s molecular interactions with STIM1, Orai1, or other proteins, providing deeper mechanistic insights. Beyond liver cancer, it is important to assess whether SARAF’s potential tumor-suppressive function is conserved in other malignancies characterized by dysregulated calcium signaling, such as breast cancer and colorectal cancer [9,26,46]. Additionally, studies should examine SARAF’s effects in normal cells to ensure therapeutic specificity and explore its role in vivo using animal models. Lastly, the design and development of small molecules or gene therapy approaches targeting SARAF expression or activity could pave the way for novel therapeutic strategies in HCC.

## 4. Materials and Methods

### 4.1. Cell Culture

Human HepG2 cells (ATCC, HB-8065) were cultured in Eagle’s Minimum Essential Medium (EMEM) supplemented with 10% (*v*/*v*) fetal bovine serum (Gibco; Billings, MT, USA) and 100 U/mL penicillin and streptomycin (Gibco; USA). The cells were maintained at 37 °C with 5% CO_2_.

### 4.2. Transfections of SARAF Overexpression

The Lipofectamine 3000 transfection protocol is widely used for the introduction of plasmid DNA into mammalian cells. The cells prepared by being seeded in a 6-well plate at a density of 1 × 10^5^ cells per well, with adherence allowed for 24 h prior to transfection. Lipofectamine 3000 used in combination with overexpression SARAF plasmid pCMV6-AC-GFP (Origene, Rockville, MD, USA, RG201864) to establish overexpression cells. For the transfection complex, 2 µg of plasmid DNA was diluted in 125 µL of Opti-MEM. In a separate tube, 3.75 µL of Lipofectamine 3000 reagent was mixed with 125 µL of Opti-MEM and incubated for 5 min at room temperature. After this, the diluted DNA was combined with the Lipofectamine 3000 mixture, with gentle mixing achieved by pipetting and allowing the complex to form for 20 min at room temperature. Once the transfection complex was ready, it was added dropwise to the cells, ensuring even distribution by gently swirling the plate. The cells were incubated at 37 °C in a CO_2_ incubator, with the transfection complex left on the cells for 6 h before the medium was replaced with fresh complete growth medium. After 24–72 h, transfection efficiency was assessed using methods of quantitative PCR (qPCR) for gene expression analysis.

### 4.3. SARAF Silencing

Silencing RNA (MISSION Predesigned siRNA) from Sigma (Kanagawa, Japan) was used in conjunction with Lipofectamine™ RNAiMAX Reagent for the transient knockdown of SARAF expression. Control siRNA used was Sigma universal non-target siRNA (SIC-001). The sequences for SARAF siRNAs were:(SASI_Hs01_00070453) CTCTTACCCTCCACTATGA(SASI_Hs02_00348914) CCTTTGTAGTCTATAAGCTCGGACTTAGATATTGCATACA

The protocol was carried out by first preparing the cells to be transfected by Reverse transfection. HepG2 cells were seeded in a 24-well plate at a density of 0.5 × 10^5^ on the day of transfection. Afterwards, 10 pmol of silencing RNA was diluted in 50 µL of Opti-MEM. In a separate tube, the Lipofectamine™ RNAiMAX Reagent was diluted in 50 µL of Opti-MEM. The diluted SARAF silencing RNA and the diluted transfection reagent were then combined and incubated for a specified time of 20 min to allow complexes to form. After the incubation period, the transfection complex was added dropwise to the cells. The cells were then incubated at 37 °C in a CO_2_ incubator for 72 h, allowing for the effective silencing of SARAF expression. Following the incubation, the expression levels of SARAF were assessed after 72 h to confirm the efficiency of the knockdown. The silence of SARAF expression was analyzed by qRT-PCR.

### 4.4. Assessment of SARAF Expression by Quantitative Real-Time Polymerase Chain Reaction (qRT-PCR)

Total RNA from all experimental samples and controls were isolated by High Pure RNA Isolation kit (Roche, Atlanta, GA, USA). Total RNA of 1 µg and a random primer was used for cDNA synthesis by High-Capacity Reverse Transcription Kit (Applied Biosystems, San Francisco, CA, USA).

The qRT-PCR was performed using Fast SYBR™ Green Master Mix (Applied Biosystems). Relative quantification of SARAF expression to housekeeping gene, glyceraldehyde-3-phosphate dehydrogenase (GAPDH), was obtained by 7500 Fast Real-Time PCR System (Applied Biosystem). The crossing point (Cp) for each sample was measured, followed by normalization of SARAF expression with GAPDH, after which the expression fold change between transfected and then non-transfected cells was calculated by the 2^−ΔΔCt^ method. Gel electrophoresis was performed on amplified PCR products to check the silencing and the overexpression of *SARAF* gene expression using GAPDH as a housekeeping control gene, then they were stained with Ethidium bromide and visualized under UV instrument.

### 4.5. Intracellular Calcium Level Measurement by SARAF Modulation

Calcium Assay Kit (Colorimetric) ab102505 was used to determine calcium concentration within the physiological range of 0.4–100 mg/dL (0.2–1 mM).

Transfected and non-transfected control cells were harvested. The cells were washed with cold phosphate-buffered saline (PBS) to remove any residual media. Subsequently, the cells were resuspended in 500 μL of Calcium Assay Buffer (PBS + 0.1% NP-40), and the suspension was placed on ice. To ensure proper homogenization, the cells were quickly homogenized by pipetting up and down several times. The sample was then centrifuged for 2 to 5 min at 4 °C at maximum speed to eliminate any insoluble material. The supernatant was carefully collected and transferred to a clean tube, where it was kept on ice until further use.

Prior to the assay, the calcium standards were diluted according to the recommended kit protocol. Reaction wells were set up by adding 50 µL of standard dilutions to the standard wells and 50 µL of samples to the sample wells. Following this, 90 µL of the Chromogenic Reagent was added to each well containing standards, samples, and controls. Additionally, 60 µL of Calcium Assay Buffer was introduced into each well. The contents of the wells were mixed thoroughly and incubated at room temperature for 10 min. Absorbance was then measured using a microplate reader at an optical density of 575 nm (OD575 nm).

### 4.6. Cell Proliferation Assay

Water Soluble Tetrazolium (WST-1) reagent was used to measure cell proliferation. HepG2 cells were seeded in 6-well plates at a density of 2 × 10^5^ cells/well for the transfection experiment. After the 72 h incubation, the HepG2 cells from the 6-well plates (both SARAF silenced/overexpression and negative control) were collected and seeded in 100 μL 96-well plates at a density of 5 × 10^3^ cells/well, and then incubated for an additional 24 h. After that, 10 μL of WST-1 reagents were added to each well and incubated for 4 h. The absorbance was measured at 570 nm using a microplate reader (BioTek Epoch2 microplate spectrophotometer, BioTek, Winooski, VT, USA). The reaction was performed in triplicate. The percentage of cell proliferation was calculated for the SARAF-silenced HepG2 cells relative to the negative control HepG2 cells. Statistical analysis was performed to determine the significance of the observed differences in cell proliferation. For the negative control, HepG2 cells were seeded in 6-well plates at the same density as the SARAF silencing experiment but without transfection. The negative control HepG2 cells were incubated in parallel with the SARAF silencing experiment.

### 4.7. Transwell Chamber Migration Assays

8 µm pore Transwell^®^ chambers for cell migration assay were used to study cell migration (R&D Systems Cell Migration Assay Kit, R&D Systems, Minnneapolis, MN, USA), involving transfected and non-transfected HepG2 cells as explained previously. After 72 h, the cells were cultured in a serum-free medium to enhance ligand binding. Following this, cells were centrifuged at 250× *g* for 10 min, washed with 1× Cell Wash Buffer, counted, and resuspended at 1 × 10^6^ cells/mL in the serum-free medium. In a 96-well plate, 50 μL of the cell suspension was added to the top chamber per well. The bottom chambers received 150 μL of cell culture medium, with or without FBS as a chemoattractant. The assembled chambers were incubated at 37 °C in a CO_2_ incubator for 24 h. After incubation, the top chambers were aspirated carefully, and each well was washed with warm (37 °C) 1× Cell Wash Buffer (200 μL per well). The bottom chambers were aspirated and washed twice with 500 μL of warm (37 °C) 1× Cell Wash Buffer. A Calcein AM/Dissociation Solution was prepared by adding 12 μL of Calcein AM Solution to 10 mL of 1× Cell Dissociation Solution and added to the bottom chambers at 100 μL per well. The chambers were then reassembled and incubated for 60 min at 37 °C. The chambers were disassembled, and the bottom chamber solutions were read at 485 nm excitation and 520 nm emission. Data were compared to controls, converting relative fluorescence units (RFU) into cell numbers using standard curve created by different dilution of cells to assess migration outcomes.

### 4.8. Statistical Analyses

The data shown corresponded to the mean ± SD or mean ± SEM of at least three independent experiments. Data were analyzed using two-tailed Mann–Whitney’s or Welch’s test to compare two conditions. For multiple comparisons, one-way analysis of variance (ANOVA) tests and Dunnett’s post-test were applied. Analyses were carried out using GraphPad Prism v8.0 (GraphPad Prism, San Diego, CA, USA).

## 5. Conclusions

In conclusion, this preliminary study identified SARAF as a potential tumor suppressor and regulator of calcium homeostasis in liver cancer. These correlational findings highlight SARAF’s possible role as a therapeutic target in HCC, but further studies are required to establish causation, validate clinical relevance, and explore therapeutic potential. By demonstrating its association with inhibition of proliferation and migration while contributing to restoring calcium balance, we highlighted SARAF’s possible role as a therapeutic target in HCC. These findings are correlational and do not establish causation, as the precise molecular mechanisms and clinical relevance of SARAF remain to be fully elucidated. These findings provide a foundation for future research aimed at validating SARAF’s role and developing SARAF-based therapeutic strategies to improve outcomes for patients with liver cancer.

## Figures and Tables

**Figure 1 ijms-26-04426-f001:**
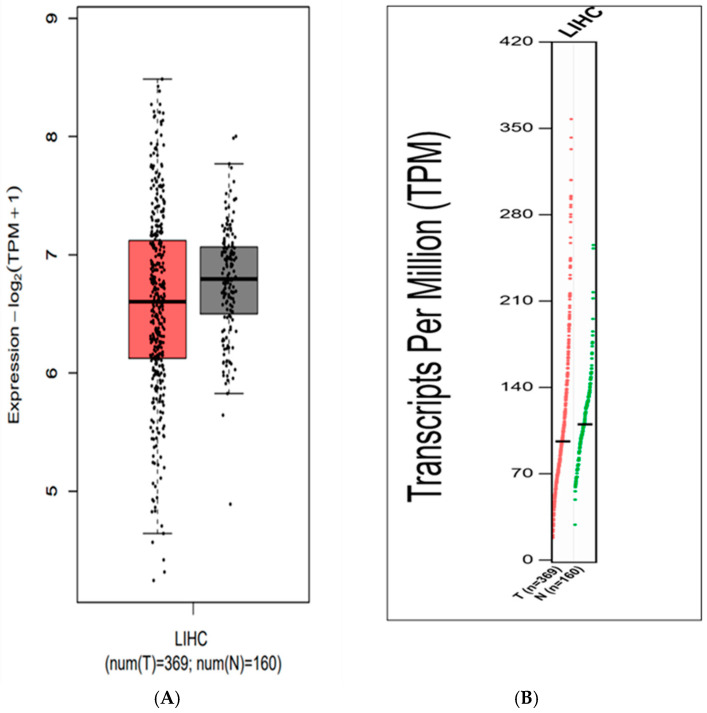
(**A**) Expression levels of the *SARAF* gene in Liver Hepatocellular Carcinoma (LIHC) samples compared to normal liver tissue, showing a trend of slightly lower expression in tumors (96.21 TPM) vs. normal samples (110.04 TPM), though not statistically significant (*p* > 0.05, GEPIA platform). (**B**) Distribution of Transcripts Per Million (TPM) values for SARAF in LIHC samples, indicating a potential downregulation in tumors that requires further validation.

**Figure 2 ijms-26-04426-f002:**
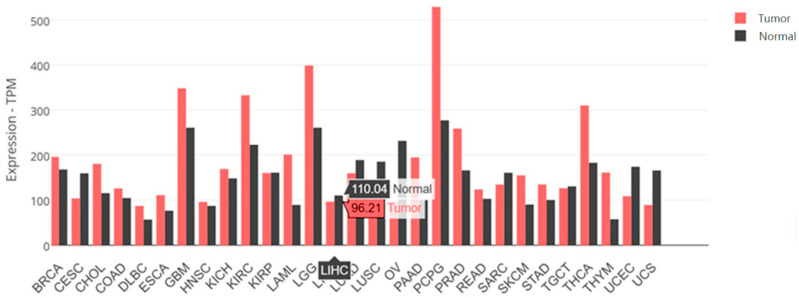
The *SARAF* gene expression data for LIHC samples. *SARAF* gene expression in tumor samples are 96.21 TPM. Normal liver samples have *SARAF* gene expression of 110.04 TPM. Also displayed are the expression levels of the *SARAF* gene in a variety of cancer and normal tissue samples.

**Figure 3 ijms-26-04426-f003:**
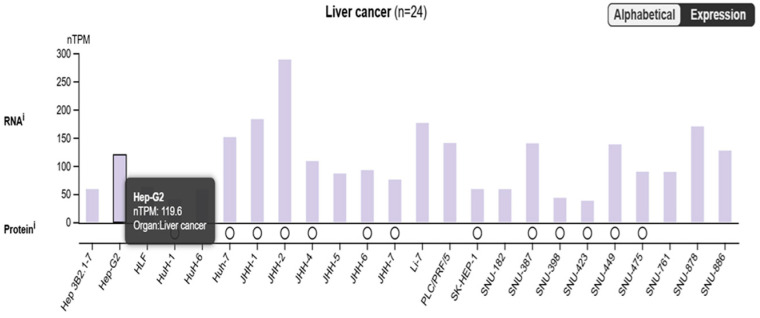
The figure highlighted the SARAF expression, which has a TPM value of 119.6 in the liver cancer samples. This suggests that the *SARAF* gene is expressed in liver cancer, with a relatively high expression level compared to other genes shown in the plot.

**Figure 4 ijms-26-04426-f004:**
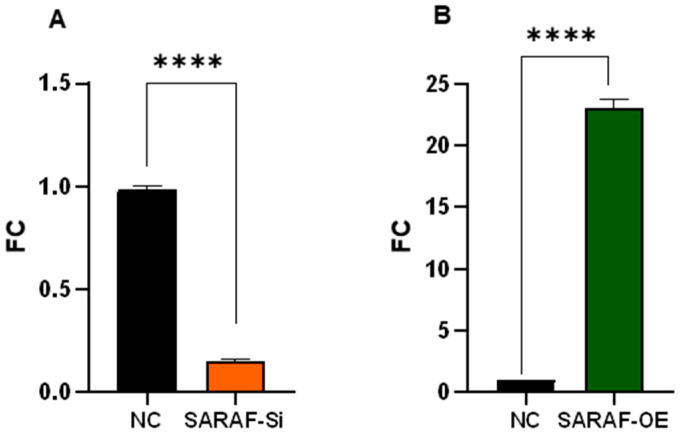
SARAF expression levels in HepG2 cells after genetic manipulation. Quantitative real-time PCR analysis of SARAF expressions in HepG2 cells following transfection with control siRNA, SARAF-specific siRNA (**A**), or SARAF overexpression plasmid (**B**). Data are presented as fold change relative to control (2^−ΔΔCt^ method). SARAF expression was significantly reduced 0.15-fold in siRNA-treated cells, while a robust 22.5-fold increase was observed in cells transfected with the plasmid overexpression compared to control (**** *p* < 0.0001).

**Figure 5 ijms-26-04426-f005:**
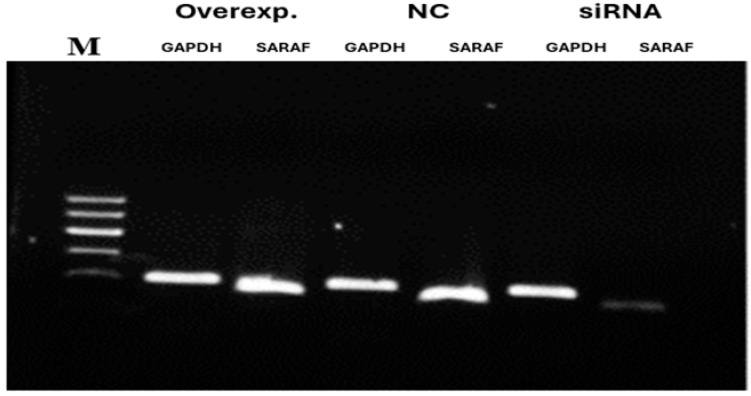
Agarose gel (1.5%) electrophoresis for mRNA expression of *SARAF* gene following silencing and overexpression. The gel shows amplified qRT-PCR products stained with Ethidium bromide and visualized under UV. Overexpression (Overexp.) displays a high-intensity band, silencing (S) shows a faint band, and negative control (N) is intermediate, with GAPDH as a housekeeping control. Note that this represents mRNA levels, not protein expression.

**Figure 6 ijms-26-04426-f006:**
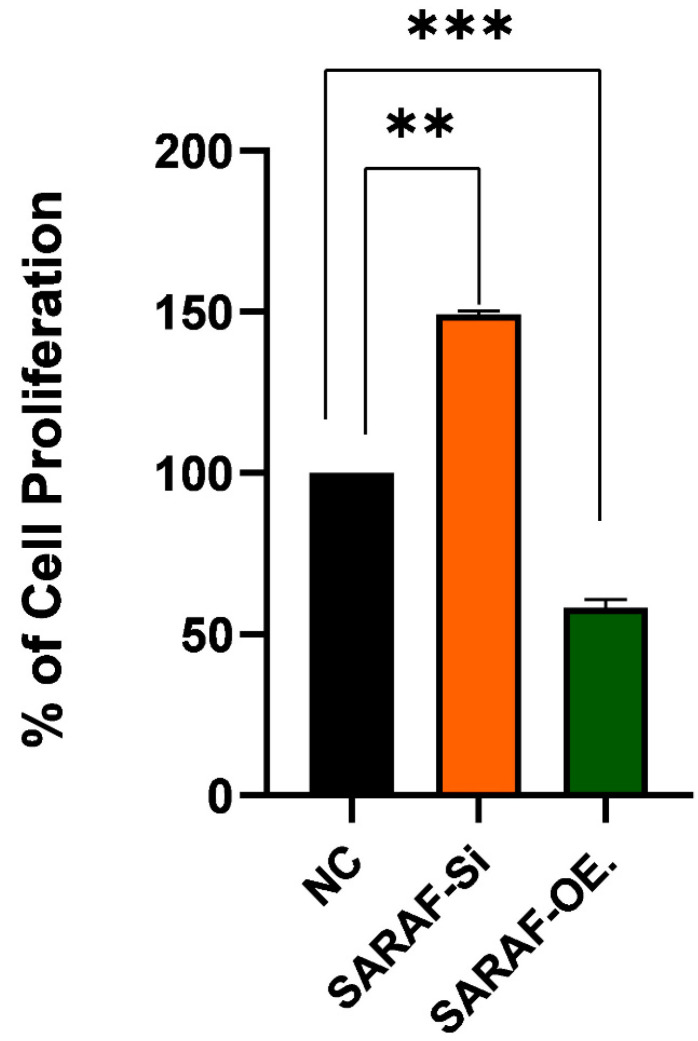
Effect of SARAF silencing and overexpression on HepG2 liver cancer cell line proliferation SARAF silencing (SARAF siRNA) significantly increases cell proliferation by approximately 50% compared to the negative control (NC), while SARAF overexpression reduces cell proliferation by approximately 50% compared to the NC. Statistical analysis using repeated measures ANOVA showed a highly significant difference between the groups (*** *p* < 0.001, ** *p* < 0.01), confirming the impact of SARAF silencing and overexpression on HepG2 cell proliferation. Data are expressed as a percentage of cell proliferations.

**Figure 7 ijms-26-04426-f007:**
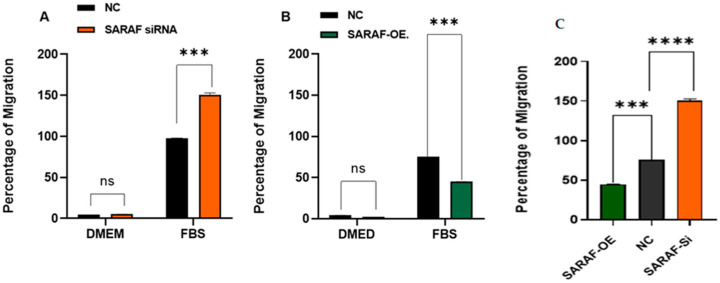
(**A**) Percentage of migration of HepG2 cells in control (NC) and SARAF knockdown (siRNA SARAF) groups under serum-free conditions (DMEM) and in response to FBS as a chemoattractant. A significant increase (*** *p* < 0.001) in migration was observed in SARAF-silenced cells compared to the control group, indicating that SARAF knockdown enhances the migratory potential of HepG2 cells. Data are presented as mean ± standard deviation. (**B**) Percentage of migration of HepG2 cells in control (NC) and SARAF overexpression groups under serum-free conditions (DMEM) and in response to FBS as a chemoattractant. A significant reduction (*** *p* < 0.001) in migration was observed in SARAF-overexpressing cells compared to the control group, suggesting that SARAF overexpression inhibits the migratory potential of HepG2 cells. Data are presented as mean ± standard deviation. (**C**) SARAF overexpression significantly reduced migration compared to SARAF silencing (*** *p* < 0.001, **** *p* < 0.0001).

**Figure 8 ijms-26-04426-f008:**
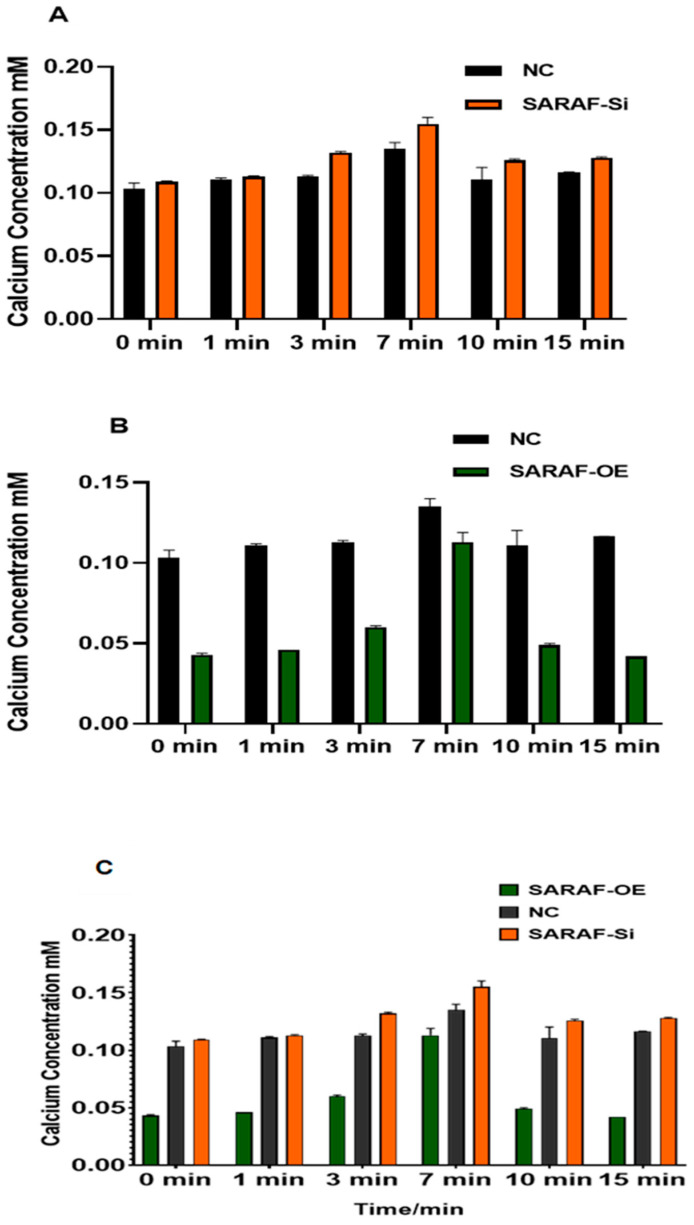
(**A**) Intracellular calcium concentration (mM) measured at six time points (0 min, 1 min, 3 min, 7 min, 10 min, and 15 min) in control and SARAF knockdown (siRNA SARAF) HepG2 cells. SARAF knockdown did not show significant differences compared to the control group, indicating no further impact on the already disrupted calcium homeostasis in HepG2 cells. Data are presented as mean ± standard deviation. (**B**) Intracellular calcium concentration (mM) measured at six time points (0 min, 1 min, 3 min, 7 min, 10 min, and 15 min) in control and SARAF-overexpressing HepG2 cells. SARAF overexpression reduced the disturbance in calcium homeostasis, with calcium levels showing a significant decrease at later time points (7 min and 10 min) compared to the control group. These findings suggest that SARAF overexpression helps mitigate the elevated calcium levels characteristic of HepG2 cells. Data are presented as mean ± standard deviation. (**C**) When comparing SARAF silencing and overexpression groups (Figure 7C), SARAF overexpression led to a reduction in calcium levels at later time points compared to controls (*p* < 0.05), but the difference between silencing and overexpression groups was not statistically significant (*p* > 0.05).

## Data Availability

Data will be available upon request.

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
