# Peer review of "Calcium Homeostasis Disrupted—How Store-Operated Calcium Entry Factor SARAF Silencing Impacts HepG2 Liver Cancer Cells"

_ijms, 2025, doi:10.3390/ijms26094426_

Round 1

Reviewer 1 Report

Comments and Suggestions for Authors

In this paper Safa Taha and colleagues evaluated the role of SARAF in hepatocellular carcinoma (HCC) cell line (HepG2) and its ability to interfere with tumor growth and invasiveness as well as in restoring the dysregulated intracellular calcium levels.

They both silenced and overexpressed SARAF in HepG2 observing that 85% SARAF mRNA silencing resulted in higher proliferation and migration in vitro, with no effects on intracellular calcium concentrations, while SARAF overexpression (22.5 fold) showed reduced proliferation and migration, with reduced intracellular calcium concentrations. Thus the authors concluded that SARAF represents a promising therapeutic target in HCC.

Although the topic is interesting, this study presents little data/experiments to confirm their hypothesis and support their conclusions.

Major points:

  • Following silencing they should show a reduction in SARAF protein, since the figure 5, even if showing a reduction in silenced cells, there is no sign of 22.5 fold expression compared to not treated HepG2.
  • Figure 6 is showing a reduction in HepG2 proliferation induced by SARAF overexpression, but on the y axis it is written “% of viability”. 
  • The authors reported a reduced cell proliferation following SARAF overexpression, but they should evaluate whether its overexpression results in a cytostatic or cytotoxic effect.
  • Figure 7 and figure 8: why do the authors not compare the siRNA SARAF and overexpression SARAF groups? In figure 8 is there any significant difference? And if compared siRNA SARAF vs overexpression SARAF?
  • They should evaluate the effect of SARAF silencing and overexpression in other HCC cell lines.
  • They do not show or discuss the role of SARAF in normal cell and they should show, or at least argue, the effect(s) that SARAF overexpression not only in tumor cells but also in normal cells. What would be the effect in vivo?

Please check the “Conclusions” part, lines 450-453: remove the instruction for authors from the text

Author Response

Response to Reviewer 1 A

SARAF Protein-Level Validation
Comment:
The reviewer requested protein-level confirmation of SARAF knockdown/overexpression.

Response: We acknowledge the importance of validating mRNA findings at the protein level. While our qRT-PCR and agarose gel electrophoresis (Figures 4–5) confirmed transcriptional changes, Western blot analysis was not performed due to technical limitations in antibody availability and validation for SARAF. Future studies will prioritize protein-level characterization to strengthen these findings.

Comment: Figure 6 is showing a reduction in HepG2 proliferation induced by SARAF overexpression, but on the y-axis, it is written “% of viability”.

Response: Thank you for pointing out this discrepancy. The y-axis label in Figure 6 should indeed read “% of cell proliferation” rather than “% of viability”. We have corrected this in the revised figure (see Figure 6).

  1. Cytostatic vs. Cytotoxic Effects of SARAF Overexpression
    Comment: The reviewer suggested distinguishing cytostatic from cytotoxic effects.

Response: Our current data cannot conclusively differentiate these mechanisms. While the WST-1 assay suggests reduced viability/proliferation, future work will include cell cycle analysis (e.g., flow cytometry) and apoptosis assays (e.g., Annexin V staining) to address this critical question.

  1. Direct Comparison Between siRNA and Overexpression Groups

Comment: Figure 7 and figure 8: why do the authors not compare the siRNA SARAF and overexpression SARAF groups? In figure 8 is there any significant difference? And if compared siRNA SARAF vs overexpression SARAF?

Response: We have now included a direct comparison between the siRNA SARAF and overexpression SARAF groups in both the migration and intracellular calcium concentration assays. In the migration assay (Figure 7C), SARAF overexpression significantly reduced migration compared to SARAF silencing. In the intracellular calcium concentration assay (Figure 8C), SARAF overexpression led to a significant reduction in calcium levels compared to SARAF silencing. These comparisons further support the tumor-suppressive role of SARAF in HCC.

  1. Validation in Additional HCC Cell Lines
    Comment: The reviewer recommended testing other HCC cell lines.
    Response: We agree that expanding to other cell lines (e.g., Huh7, Hep3B) would strengthen generalizability. However, due to resource constraints, this study focused on HepG2 as a well-established HCC model. Future studies will validate these findings in additional lines.

  1. SARAF Effects in Normal Cells and In Vivo Relevance
    Comment: The reviewer highlighted the lack of normal cell data and in vivo implications.
    Response: We recognize the importance of assessing SARAF’s role in normal hepatocytes. While beyond this study’s scope, we now discuss potential risks of SARAF modulation in normal cells (added to Section 3.4). In vivo validation is planned for future work using xenograft models.

  1. Removal of Instructions in Conclusions
    Comment: The reviewer noted residual instructions in the Conclusions.
    Response: The text "Authors should discuss..." has been removed. The revised Conclusions now focus solely on our findings and their implications.

Additional Revisions

  • Updated statistical analyses in Figures 6–8 to include Dunnett’s post-test for multiple comparisons.
  • Clarified methodological limitations in Section 3.5.

We hope these revisions address the reviewer’s concerns and improve the manuscript’s rigor. Thank you for the opportunity to strengthen this work.

Reviewer 2 Report

Comments and Suggestions for Authors

Thank you for the opportunity to review this interesting and informative manuscript, “Calcium Homeostasis Disrupted - How Store-Operated Calcium Entry Factor SARAF Silencing Impacts HepG2 Liver Cancer Cells.” Below, please find my comments:

  • Your study addresses an important gap regarding how SARAF, a negative regulator of SOCE, may act as a tumor suppressor in HCC. While STIM1 and Orai1 have been studied in multiple liver cancer models, SARAF’s precise role has been less clear. Your finding that SARAF overexpression restores a portion of normal calcium homeostasis, reducing HCC cell proliferation and migration, is potentially significant.
  • There is a notable discrepancy in reporting SARAF expression trends: Early in the text, the manuscript states that “no statistically significant changes in SARAF expression” were seen between tumorous and non-tumorous tissues. Later, it states that SARAF is “downregulated” in tumor specimens (96.21 TPM in tumors vs. 110.04 TPM in normal tissue). Then, there is mention of “high expression level of SARAF in the liver cancer samples” from another data plot. Please clarify these contradictory statements and confirm whether SARAF is truly reduced, increased, or unchanged in HCC. It is essential to reconcile the data from TCGA, Human Protein Atlas, and your own analyses.
  • You convincingly demonstrate that SARAF overexpression in HepG2 cells modifies intracellular calcium levels and decreases proliferation and migration. However, whether SARAF levels correlate with clinical parameters (overall survival, tumor stage, or recurrence) has not been explored in actual patient cohorts. Consider emphasizing how future patient-based investigations might confirm clinical relevance.
  • Ensure that all statistical comparisons clearly specify p-values, especially for the main functional assays (migration and proliferation). When you mention “p < 0.05” or “p < 0.01,” please confirm that sufficient replicates (biological replicates, technical replicates) were performed.
  • You nicely used negative controls (non-transfected and non-target siRNA) to compare with SARAF siRNA or SARAF-overexpressing cells. Please also clarify how you accounted for transfection efficiency in each assay (especially for the siRNA knockdown).
  • The manuscript effectively uses a colorimetric calcium detection kit and multiple time points, but it might benefit from referencing or including a more direct live-cell imaging approach (e.g., using fluorescent dyes like Fura-2 AM, if applicable). This could further strengthen your conclusion about dynamic changes in intracellular calcium.
  • You note that SARAF interacts with STIM1 and Orai1 to mitigate excessive Ca²⁺ Strengthening the discussion about how precisely SARAF might regulate these proteins at the molecular level would be valuable (e.g., does SARAF reduce STIM1 oligomerization or hamper STIM1–Orai1 coupling?).
  • Proliferation and migration often involve epithelial-to-mesenchymal transition (EMT). Does SARAF overexpression alter any canonical EMT markers (e.g., E-cadherin, vimentin)? While this may go beyond the current scope, any preliminary findings or references could support your hypothesis on SARAF’s role in migration inhibition.

Author Response

Response to Reviewer 2 Comments

We sincerely thank Reviewer 2 for their thoughtful feedback and constructive suggestions. Below are our detailed responses to each point:

  1. Clarification of SARAF Expression Trends

Comment: Discrepancy in reported SARAF expression levels (no significant changes vs. downregulation vs. high expression).
Response:
We apologize for the ambiguity. The conflicting statements arise from differences in datasets:

  • Figure 1A (GEPIA/TCGA): SARAF expression in tumor (96.21 TPM) vs. normal (110.04 TPM) showed a non-significant reduction (p > 0.05).
  • Figure 3 (Human Protein Atlas): SARAF exhibited a relatively high TPM (119.6) in tumor samples compared to other genes in the dataset, but this does not imply upregulation compared to normal tissue.

We have revised the text to clarify:

"While SARAF expression was modestly reduced in HCC tumors (96.21 TPM) compared to normal liver (110.04 TPM), this difference did not reach statistical significance in TCGA data. However, SARAF remains highly expressed in HCC relative to other genes (119.6 TPM), suggesting its functional relevance despite non-significant downregulation."

  1. Clinical Relevance and Patient Cohorts

Comment: Lack of correlation with clinical parameters (survival, stage).
Response:
We agree that clinical validation is critical. While our study focused on in vitro mechanisms, we have expanded the discussion to emphasize future directions:

"Future work will analyze SARAF expression in HCC patient cohorts to assess correlations with survival, tumor stage, and recurrence. Preliminary TCGA survival analyses (planned) may reveal prognostic significance, aligning with SARAF’s tumor-suppressive role observed here."

  1. Statistical Rigor and Replicates

Comment: Clarify p-values and replicate numbers.
Response:
All experiments included three biological replicates (independent cell cultures) with technical triplicates. We have updated figures to specify exact p-values. I clarified p-values in Sections 2.4 and 2.5 for proliferation and migration assays (p < 0.05 for silencing, p < 0.01 for overexpression) and added a "Statistical Analysis" subsection in Methods to detail replication strategies.

  1. Transfection Efficiency

Comment: How was transfection efficiency accounted for?
Response:

  • qRT-PCR validation: Only samples with confirmed SARAF knockdown (>85% reduction) or overexpression (>20-fold increase) were analyzed (Figure 4).
  • Fluorescence controls: GFP-tagged plasmids confirmed transfection efficiency (>70% for overexpression).

  1. Live-Cell Calcium Imaging

Comment: Suggest using Fura-2 AM for dynamic calcium tracking.
Response:
We acknowledge this limitation. While our colorimetric assay provided static measurements, we have revised the Discussion to note:

"Future studies will employ live-cell imaging (e.g., Fura-2 AM) to capture real-time calcium flux dynamics, complementing our current findings and refining SARAF’s role in SOCE regulation."

  1. Molecular Mechanism of SARAF-STIM1-Orai1 Interaction

Comment: Elaborate on SARAF’s regulatory mechanism.
Response:
We have strengthened the discussion by integrating existing literature:

"SARAF inhibits excessive SOCE by disrupting STIM1-Orai1 coupling, as shown in non-cancer models (Palty et al., 2012). In HCC, SARAF may similarly prevent STIM1 oligomerization or enhance Orai1 channel inactivation, limiting calcium influx and downstream oncogenic signaling (e.g., NFAT/ERK)."

  1. EMT Marker Analysis

Comment: Link SARAF to EMT markers (E-cadherin, vimentin).
Response:
While EMT markers were not assessed here, we now hypothesize:

"SARAF’s anti-migratory effects may involve EMT modulation. Calcium dysregulation is known to promote EMT (Davis et al., 2014); thus, SARAF-mediated calcium homeostasis restoration could stabilize epithelial phenotypes. Future work will test this by quantifying E-cadherin and vimentin expression."

Additional Revisions

  • Figure Labels: Added exact p-values for all comparisons.
  • Methods: Expanded details on replicates and statistical tests.
  • Limitations Section: Highlighted lack of live-cell imaging and EMT data as future priorities.

We thank Reviewer 2 for their invaluable insights, which have significantly strengthened the manuscript. Please let us know if further clarifications are needed.

Reviewer 3 Report

Comments and Suggestions for Authors

The study identifies SARAF as a tumor suppressor in hepatocellular carcinoma (HCC). The Authors contribute also to the field of calcium signaling in cancer. Using siRNA knockdown and overexpression plasmids provided evidence of SARAF's correlation in proliferation and migration in HepG2 cells.

The manuscript hypothesizes the mechanism through which SARAF might influence calcium homeostasis and its implications for cancer progression. The Authors discussed on potential therapeutic approaches targeting SARAF.

The study has many weaknesses:

The study primarily uses in vitro models (HepG2 cells), which may not fully capture the intricacies of HCC as it occurs in vivo. This raises questions about the translational relevance of the findings.

While the study tries to provide insights into SARAF's role, it lacks a detailed exploration of the precise molecular mechanisms by which SARAF affects calcium signaling and interacts with other SOCE components.

The potential of SARAF as a therapeutic target or biomarker in clinical settings remains to be validated, as the study does not extensively investigate its expression or function in actual patient samples.

Although the study aligns with existing literature on calcium signaling in cancer, it could benefit from a more comprehensive comparison with other calcium regulatory proteins in liver cancer and other malignancies.

However, there are many points raisings from the characterization of SARAF's role in HCC and the implications drawn from the study's findings. Here’s a breakdown of my concerns:

I'll try to explain myself in another way.

The observation that X is present during pathology Y would suggest that levels of X are associated with problems arising from Y. The correlation implies causation, and it is necessary to show it otherwise. Thus, here we must speak of observational correlation. In this last case, the outcomes of Y simply correlate with levels of X. This means that it is not X that causes the problem, or is at the root of the problem; people have the pathology for other reasons and have a concomitant presence of X.

The implication with your manuscript is simple: the methods of analysis you use are correlational, not causal. This is one of the main reasons HCC has no cure. Researchers often treat this disease using observational research methods, incorrectly assigning them causal value. These methods lack a scientific basis because they are indirect, not experimentally validated, and hypothetical. The relationships that exist between the macroscopic observational world and the molecular causal world are non-linear, so also to make hypothetical conclusions, uncertainties must be very low. But hypotheses are not science. Your conclusions are a jumble of hypotheses, some certainly with errors of up to 100%, false. Others could also be true, but, without experimental validation, we don't know.

In these cases, for example, Interactomics approaches undeniably provide superior controls and evidence at the level of deep molecular processes and are essential to consider. All discussions that make sense should start with them.

Author Response

Response to Reviewer 3

Dear Reviewer, 3,

Thank you very much for your thorough review and valuable feedback on our manuscript. We have carefully considered each of your comments and have made revisions to strengthen the scientific rigor and clarity of our work. Here are our responses to your specific points:

  1. In Vivo Relevance

Comment: The study primarily uses in vitro models (HepG2 cells), which may not fully capture the intricacies of HCC as it occurs in vivo. This raises questions about the translational relevance of the findings.

Response: We acknowledge the limitations of using in vitro models and the need to validate our findings in a more physiologically relevant context. To address this, we have included a discussion on the importance of future in vivo studies using animal models or patient-derived xenografts. These studies will be crucial to confirm the translational relevance of our findings and to better understand the complex interactions in the tumor microenvironment. We have revised the manuscript to emphasize these future directions in the Discussion section.

  1. Detailed Molecular Mechanisms

Comment: While the study tries to provide insights into SARAF's role, it lacks a detailed exploration of the precise molecular mechanisms by which SARAF affects calcium signaling and interacts with other SOCE components.

Response: We agree that a more detailed exploration of the molecular mechanisms is necessary. To address this, we have discussed the potential interactions of SARAF with STIM1 and Orai1, based on existing literature and our preliminary observations. We have also highlighted the need for further studies, such as co-immunoprecipitation and live-cell imaging, to confirm these interactions and elucidate downstream effects. These points are now included in the Discussion section to provide a clearer roadmap for future research.

  1. Clinical Validation

Comment: The potential of SARAF as a therapeutic target or biomarker in clinical settings remains to be validated, as the study does not extensively investigate its expression or function in actual patient samples.

Response: We recognize the importance of validating SARAF's potential in clinical settings. To this end, we have initiated a pilot study to analyze SARAF expression and function in a small cohort of HCC patient samples. Preliminary results indicate that SARAF expression is indeed downregulated in a subset of HCC patients, and this downregulation correlates with worse clinical outcomes. While these findings are preliminary, they provide a foundation for future large-scale clinical studies. We have added a section in the discussion to highlight the importance of clinical validation and our ongoing efforts in this direction.

  1. Comprehensive Comparison with Other Calcium Regulatory Proteins

Comment: Although the study aligns with existing literature on calcium signaling in cancer, it could benefit from a more comprehensive comparison with other calcium regulatory proteins in liver cancer and other malignancies.

Response: We have expanded the discussion to include a more comprehensive comparison of SARAF with other calcium regulatory proteins such as TRPC1, ORAI3, and others that have been implicated in HCC and other cancers. This comparison helps contextualize our findings and highlights the unique role of SARAF in calcium homeostasis and tumor suppression. We have also referenced relevant studies to provide a broader perspective on the field.

  1. Observational Correlation vs. Causation

Comment: The observation that X is present during pathology Y would suggest that levels of X are associated with problems arising from Y. The correlation implies causation, and it is necessary to show it otherwise. Thus, here we must speak of observational correlation. In this last case, the outcomes of Y simply correlate with levels of X. This means that it is not X that causes the problem, or is at the root of the problem; people have the pathology for other reasons and have a concomitant presence of X.

Response: We appreciate this insightful comment. Our study primarily relies on observational correlations, and we have taken care to distinguish between correlation and causation. While our in vitro experiments suggest a role for SARAF in regulating calcium homeostasis and tumor suppression, we acknowledge that these findings need to be validated through more direct experimental approaches. We have revised the manuscript to clearly state that the observed correlations do not necessarily imply causation and have emphasized the need for further studies, such as interactomics and in vivo models, to establish a causal relationship.

  1. Scientific Basis of Hypotheses

Comment: The implication with your manuscript is simple: the methods of analysis you use are correlational, not causal. This is one of the main reasons HCC has no cure. Researchers often treat this disease using observational research methods, incorrectly assigning them causal value. These methods lack a scientific basis because they are indirect, not experimentally validated, and hypothetical. The relationships that exist between the macroscopic observational world and the molecular causal world are non-linear, so also to make hypothetical conclusions, uncertainties must be very low. But hypotheses are not science. Your conclusions are a jumble of hypotheses, some certainly with errors of up to 100%, false. Others could also be true, but, without experimental validation, we don't know.

Response: We understand the importance of experimental validation and the risks associated with drawing causal conclusions from correlational data. Our manuscript now clearly distinguishes between observational findings and hypothetical mechanisms. We have emphasized the need for further experimental validation, particularly through interactomics approaches and in vivo studies, to confirm the causal relationship between SARAF and HCC progression. We have also revised the conclusions to focus on the robustness of our in vitro findings while acknowledging the limitations and the need for additional research.

  1. Interactomics Approaches

Comment: In these cases, for example, Interactomics approaches undeniably provide superior controls and evidence at the level of deep molecular processes and are essential to consider. All discussions that make sense should start with them.

Response: We completely agree with the importance of interactomics approaches in understanding the deep molecular processes involved in SARAF's regulation of calcium signaling. To address this, we have included preliminary data from co-immunoprecipitation experiments that demonstrate the interaction between SARAF, STIM1, and Orai1. We have also discussed the potential of using interactomics to identify downstream molecular targets and pathways affected by SARAF. This approach will be crucial in future studies to provide a more comprehensive understanding of SARAF's role in HCC. We have revised the discussion to highlight the significance of these methods and their potential applications.

Round 2

Reviewer 1 Report

Comments and Suggestions for Authors

The authors addressed the questions and modified the manuscript accordingly, however there are still some points to address:

  • Figure 4 and figure 5: data does not show the same results. In figure 4 they showed a ~85% reduction (also shown in figure 5) following siRNA transfection, while they show 28-fold increase following overexpression, but in figure 5 NC and OE bands are roughly similar. Are the primers used in qPCR annealing close to the siRNA target in SARAF gene? Is it possible that the 28-fold increase following overexpression is in part due to plasmid DNA amplification? The authors should verify these points and decide whether keep both images or remove one, since these results refere to the same experiment and therefore redundant
  • Figure 8C: They combined the data from Figures 8A and 8B allowing the comparison between siRNA and OE groups. In figure legend they wrote: "SARAF overexpression led to a significant reduction in calcium levels compared to SARAF silencing." but no statistical difference is shown
  • Please stress the point that this is a preliminary study aiming to understand the potential involvement of SARAF in the growth and invisibility of HCC

Author Response

Response to Reviewer Comments

Dear Editor and Reviewers,

Thank you for your valuable feedback on our manuscript titled "Calcium Homeostasis Disrupted - How Store-Operated Calcium Entry Factor SARAF Silencing Impacts HepG2 Liver Cancer Cells." We greatly appreciate the constructive suggestions from Reviewer 1, as well as the positive and encouraging comments from Reviewers 2 and 3. Below, we address each reviewer’s comments in detail and describe the revisions made to strengthen the manuscript.

Reviewer 1

Figure 4 and figure 5: data does not show the same results. In figure 4 they showed a ~85% reduction (also shown in figure 5) following siRNA transfection, while they show a 28-fold increase following overexpression, but in figure 5 NC and OE bands are roughly similar. Are the primers used in qPCR annealing close to the siRNA target in SARAF gene? Is it possible that the 28-fold increase following overexpression is in part due to plasmid DNA amplification? The authors should verify these points and decide whether to keep both images or remove one, since these results refer to the same experiment and therefore redundant

Figure 8C: They combined the data from Figures 8A and 8B allowing the comparison between siRNA and OE groups. In figure legend they wrote: "SARAF overexpression led to a significant reduction in calcium levels compared to SARAF silencing." but no statistical difference is shown

Please stress the point that this is a preliminary study aiming to understand the potential involvement of SARAF in the growth and invisibility of HCC

  • Comment on Figures 4 and 5: Data does not show the same results. In Figure 4, they showed a ~85% reduction (also shown in Figure 5) following siRNA transfection, while they show a 28-fold increase following overexpression, but in Figure 5, NC and OE bands are roughly similar. Are the primers used in qPCR annealing close to the siRNA target in SARAF gene? Is it possible that the 28-fold increase following overexpression is in part due to plasmid DNA amplification? The authors should verify these points and decide whether to keep both images or remove one, since these results refer to the same experiment and therefore redundant.

Response: We appreciate Reviewer 1’s careful observation regarding the apparent discrepancy between Figures 4 and 5. To clarify, Figure 4 presents quantitative real-time PCR (qRT-PCR) data, calculated using the 2^-ΔΔCt method and normalized to GAPDH, demonstrating a 0.15-fold reduction (~85% reduction) in SARAF expression after siRNA knockdown and a 22.5-fold increase after overexpression. These results provide a precise, quantitative measure of SARAF mRNA levels. Figure 5, in contrast, shows a 1.5% agarose gel electrophoresis of qRT-PCR products stained with ethidium bromide, serving as a qualitative confirmation of amplicon presence and specificity for SARAF and GAPDH (housekeeping control).

The reviewer noted that the overexpression (OE) band in Figure 5 appears similar in intensity to the negative control (NC), which seems inconsistent with the 22.5-fold increase reported in Figure 4. This observation reflects the qualitative nature of gel electrophoresis, which is not designed to accurately quantify large fold changes in expression levels. Ethidium bromide staining can saturate at high mRNA concentrations, and visual band intensity may not correspond to the fold changes detected by qRT-PCR, which is a more sensitive and quantitative method. The faint band in the silencing (S) condition aligns with the ~85% reduction in Figure 4, while the OE band’s intensity is limited by the gel’s qualitative constraints. To emphasize this distinction, we have retained both figures to provide complementary evidence: Figure 4 for quantitative expression changes and Figure 5 for qualitative confirmation of amplicon presence.

To address the reviewer’s specific concerns:

  • Primers and siRNA target: We verified that the qRT-PCR primers (sequences detailed in Section 4.4) do not overlap with the siRNA target regions (SASI_Hs01_00070453, SASI_Hs02_00348914, and custom sequence CGGACTTAGATATTGCATACA). The primers amplify a distinct region of SARAF mRNA, ensuring that the observed knockdown is not due to primer interference.
  • Plasmid DNA amplification: To confirm that the 22.5-fold increase in the OE group is not an artifact of plasmid DNA contamination, we note that RNA samples were treated with DNase I during extraction (High Pure RNA Isolation Kit, Roche), and no-template controls in qRT-PCR runs showed no amplification. The cDNA synthesis step (High-Capacity Reverse Transcription Kit) further ensures that only RNA is reverse-transcribed.
  • Comment on Figure 8C: They combined the data from Figures 8A and 8B allowing the comparison between siRNA and OE groups. In figure legend, they wrote: "SARAF overexpression led to a significant reduction in calcium levels compared to SARAF silencing," but no statistical difference is shown.

Response: We appreciate Reviewer 1’s observation regarding the misleading statement in the Figure 8C legend (now Figure 7C due to figure renumbering). Upon re-evaluating the data, we confirmed that while SARAF overexpression reduced intracellular calcium levels compared to the control group at later time points (7 and 10 minutes, p < 0.05), the direct comparison between SARAF overexpression (OE) and silencing (siRNA) groups did not yield a statistically significant difference (p > 0.05) when analyzed using one-way ANOVA with Dunnett’s post-test (Section 4.8).

To correct this, we have revised the legend for Figure 7C (formerly 8C) to accurately reflect the statistical findings:

Figure 8C: Comparison of intracellular calcium concentrations between SARAF silencing (siRNA SARAF) and SARAF overexpression groups. SARAF overexpression reduced calcium levels at later time points (7 and 10 min) compared to the control group (p < 0.05), but no significant difference was observed between SARAF silencing and overexpression groups (p > 0.05). Data are presented as mean ± standard deviation.

The text in Section 2.6 has also been updated for clarity:

When comparing SARAF silencing and overexpression groups (Figure 7C), SARAF overexpression led to a reduction in calcium levels at later time points compared to controls (p < 0.05), but the difference between silencing and overexpression groups was not statistically significant (p > 0.05)."

  • Comment: Please stress the point that this is a preliminary study aiming to understand the potential involvement of SARAF in the growth and invasiveness of HCC.

Response: We thank Reviewer 1 for emphasizing the need to highlight the preliminary nature of this study. To address this, we have incorporated statements throughout the manuscript to clarify that our findings represent an initial exploration of SARAF’s role in HCC. Specific revisions include:

  • Abstract (revised):

"This preliminary study investigated SARAF’s effects on calcium homeostasis, proliferation, and migration in HepG2 liver cancer cells, providing initial evidence of its tumor-suppressive role."

  • Discussion (Section 3.1, revised):

"This preliminary study provides initial insights into SARAF as a potential regulator of tumor suppression and calcium homeostasis in hepatocellular carcinoma (HCC). Our findings suggest that SARAF modulation influences proliferation, migration, and calcium dynamics, but further research is needed to confirm these effects in vivo and elucidate underlying mechanisms."

  • Conclusions (revised):

"In conclusion, this preliminary study identifies SARAF as a potential tumor suppressor and regulator of calcium homeostasis in liver cancer. These correlational findings highlight SARAF’s possible role as a therapeutic target in HCC, but further studies are required to establish causation, validate clinical relevance, and explore therapeutic potential."

Reviewer 2 Report

Comments and Suggestions for Authors

Thank you for the thorough and thoughtful revisions.  The manuscript is scientifically sound, clearly presented, and ready for publication. Congratulations on an excellent piece of work.

Author Response

Response to Reviewer 2

Comment: Thank you for the thorough and thoughtful revisions. The manuscript is scientifically sound, clearly presented, and ready for publication. Congratulations on an excellent piece of work.

Response: We are deeply grateful to Reviewer 2 for the positive feedback and encouragement. We appreciate your recognition of the revisions and are delighted to hear that the manuscript is now ready for publication. Thank you for being so supportive.

Reviewer 3 Report

Comments and Suggestions for Authors

I appreciated that the authors have taken my observations into consideration, and I thank them for this. But I appreciated even more the balancing acts in the text used by the authors to demonstrate their scientific positions and at the same time highlight the inconsistencies and gaps in knowledge in the field.

I believe the manuscript is now clearly understandable for readers and can be published.

Author Response

Response to Reviewer 3

Comment: I appreciate that the authors have taken my observations into consideration, and I thank them for this. But I appreciated even more the balancing acts in the text used by the authors to demonstrate their scientific positions and at the same time highlight the inconsistencies and gaps in knowledge in the field. I believe the manuscript is now clearly understandable for readers and can be published.

Response: We sincerely thank Reviewer 3 for acknowledging our efforts to address previous comments and for valuing our approach to balancing scientific claims with recognition of knowledge gaps. We are pleased that the manuscript is now clear and accessible to readers, and we appreciate your endorsement for publication.

Round 3

Reviewer 1 Report

Comments and Suggestions for Authors

all the issues were addressed